# Assessment of Psychosocial Factors in Brazilian Dentists Facing Risk of SARS-CoV-2 Infection in the Public and Private Sectors

**DOI:** 10.3390/ijerph191710576

**Published:** 2022-08-25

**Authors:** Rogério Bertevello, Ida Regina Tomaz Carvalho da Silva Capela, Marcelo Salmazo Castro, Ana Virgínia Santana Sampaio Castilho, Ana Carolina da Silva Pinto, Gabriela de Figueiredo Meira, Silvia Helena de Carvalho Sales Peres

**Affiliations:** Department of Pediatric Dentistry, Orthodontics and Public Health, Bauru Dental School, University of São Paulo, Bauru 17012-901, São Paulo, Brazil

**Keywords:** COVID-19, dentistry, public health

## Abstract

Dentists are at high risk of SARS-CoV-2 infection due to their close proximity to patients. Thus, the fear of contamination or spreading the virus to family members, coupled with financial need, can lead professionals to experience significant overload and psychological suffering. We investigated the perceptions of dental professionals in the public and private sectors regarding fear and anxiety related to patient care and the risk of infection in the face of the COVID-19 pandemic. Based on the previous literature, we interviewed 302 dentists online using sociodemographic and professional questions. Among the professionals evaluated, 80.8% had suspended their activities for some time, 74.8% were afraid of infection at work, 86.1% feared transmitting the virus to their families, 30.1% had already been infected, 54% felt afraid when they heard the news of death caused by SARS-CoV-2, and 63.9% reported having the protective knowledge necessary to avoid infection. Dentists who worked only in the public sector and those who worked in both sectors were more afraid of being infected than professionals who worked only in private offices. Our results highlight the impact of the COVID-19 pandemic on emotional health in dentists. This study highlights the need for more support in the psychosocial field to enable dentists to overcome difficulties and maintain the provision of good dental care for the population. Continuing education should update professions with the requisite scientific and clinical knowledge to face the pandemic and achieve greater reflection on their role within this new context to improve their professional and emotional performance.

## 1. Introduction

In December 2020, cases of a severe acute respiratory infection (SARS-CoV-2; hereafter, COVID-19) caused by a new strain of coronavirus were detected in Wuhan city, Hubei Province in China [1]. In March 2020, the World Health Organization (WHO) declared a state of pandemic caused by the exponential spread of COVID-19 across five continents [2].

The clinical features caused by COVID-19 infection range from asymptomatic patients to cases of severe acute respiratory failure, with the possibility of evolving into acute respiratory distress syndrome, which can lead to patient mortality [3].

In Brazil, by April 2021, 13,445,006 cases had been registered, and 361,334 infected people had lost their lives as a result of COVID-19 [3]. According to the WHO, these numbers made Brazil the country with the second highest number of cases and deaths from COVID-19 in the world [3].

Since the original Wuhan strain, several variants of SARS-CoV-2 have been detected, mainly with mutations in the spike protein, which has led to a greater capacity of transmissibility or virulence in so-called “variants of concern.” This denotes the need for constant surveillance of viral genetic changes capable of reducing vaccine efficacy, changes in therapies, and exploration of ways to prevent contagion and viral transmission [4].

According to Peng and Huang (2020), the main route of dissemination and infection of COVID-19 is through contaminated saliva droplets [5,6].

Healthcare workers are identified as having increased potential risk of exposure to COVID-19 due to constant exposure to droplets and aerosols; specifically, professionals that need to work in urgent situations in the oral cavity [1,7]. Dentists, in particular, due to engaging in close professional–patient contact, need for patients to remove their masks to access the oral cavity. Thus, they must contend with the presence of bodily fluids and the use of aerosols that provide a propensity for particles containing pathogenic microorganisms to be spread, transforming the dental environment into an area of high potential contamination.

The aerosols generated by dental equipment may remain suspended in the air for approximately 6 h and may be inhaled by the work team and patients. In addition, they can be deposited on clinical surfaces, creating new sources of transmission and cross contamination [8].

The new work scenario generated by the COVID-19 pandemic presents a new challenge for dentists because it necessitates consideration of work team and patients’ protection measures in addition to their important role in maintaining the health and dental treatment needs of the population.

Parallel to this context, there is the emotional pressure of the fear of infection during their professional practice, the risk of transmitting COVID-19 to family members and patients, the fear of resuming treatment, financial concerns, and the need to acquire information about the virus and related protective measures. All these new issues associated with the pandemic have been associated with a greater burden on, and psychological suffering of, dental professionals [9,10]. In addition to the increased pressure in their work environment generated by the pandemic, the need for social isolation and contact with family ties and leisure routines, which is inherent in daily life, is not being fulfilled, which further burdens professionals who are already psychologically overloaded.

For the aforementioned reasons, there is a possibility that this pandemic will negatively impact Brazilian dentists, especially in terms of self-employed professionals whose earnings have been affected by the increased costs in biosecurity and the decrease in elective care. Therefore, studies that corroborate the challenges facing professional dentists during the pandemic are of extreme relevance.

Accordingly, the present study aimed to investigate the perceptions of dental professionals working in the public and private sectors regarding fear and anxiety in patient care and the risk of infection in the context of the COVID-19 pandemic.

## 2. Materials and Methods

This work followed the STROBE guidelines for cross-sectional studies [11].

### 2.1. Sample

The target population of the present study was chosen because they are health professionals who have been dealing directly with oral cavity care in an emergency situation during the COVID-19 pandemic.

The sample size calculation was performed using the number of practicing dentists in Brazil. The quantity was estimated using a prevalence of 50% for the variables related to the outcome, with a 5% margin of error and a 95% confidence interval (CI). The sample was then increased by 20%, assuming a refusal rate of 10% plus a margin of 10% to control for confounding factors. The estimated sample size was 88 dentists. The sample was composed of dentists who answered the survey questionnaire. Invitations to participate were made available on the researchers’ social networks through a link requesting that participants fill out a form containing the proposed questionnaire (n = 302). Data collection occurred in the months of September and November 2020.

### 2.2. Eligibility Criteria

#### 2.2.1. Inclusion Criteria

The inclusion criterion was to be a dentist registered with the Regional Councils of Dentistry and currently working in a public or private dental clinic or office.

#### 2.2.2. Exclusion Criteria

The exclusion criterion was questionnaires containing unanswered questions.

### 2.3. Study Variables

Through the questionnaire, information was collected on the sociodemographic conditions, e.g., gender and age, as well as professional variables such as the institution where they studied dentistry, the time of graduation, clinical specialty, and clinical workload. The question for the variable related to perceptions of COVID-19 was “Are you afraid of being infected with COVID-19 while performing a dental procedure?”

Participants also provided their perceptions and fears related to their professional performance during the COVID-19 pandemic. Questions included: “Are you afraid of becoming contaminated during your work and transmitting the virus to your family members?”; “Are you afraid when you hear that people are dying from COVID-19?”; and “Do you think that you have the necessary protective knowledge regarding COVID-19?”

The dependent variable of the present study was the fear of becoming infected by COVID-19, which had “yes” and “no” as response options. The questions were formulated based on previous studies [7,9] and the reliability and factor analysis were considered satisfactory for the sample studied.

### 2.4. Statistical Analysis

Descriptive analysis was performed using STATA 14 software (StataCorp. 2014. Stata Statistical Software: Version 14.1. College Station, TX, USA: StataCorp LP).

For the descriptive analysis, frequencies of the general characteristics of the sample and perception of coping with the COVID-19 pandemic were estimated. Multivariate logistic regression was used to estimate the association between type of service in dentistry (the independent variable) with fear of becoming infected with COVID-19 in the practice of the profession (the dependent variable). Unadjusted analyses provided preliminary associations between the predictor and outcome. Variables with *p* ≤ 0.20 were included in the adjusted analysis. The results are presented as the odds ratio (OR) and their respective 95% CI.

## 3. Results

The sample characteristics distributions are presented in Table 1.

### 3.1. Perception on How to Face the COVID-19 Pandemic

In this questionnaire, 244 (80.8%) dentists had suspended their activities for some time, and 205 (67.9%) had already returned to their normal activities during the period in which the answers were collected. Regarding the fear of being infected at work, 76 (25.2%) reported not being afraid of infection and 21 (7%) of the professionals stated that they were not postponing appointments for patients with suspicious symptoms. A total of 260 (86.1%) interviewees were afraid of becoming infected and transmitting the virus to their family members and 163 (54%) were afraid when they heard reports of deaths caused by COVID-19. Regarding the necessary knowledge of prevention, 109 (36.1%) reported not having the protective knowledge concerning viral infection (Table 2).

### 3.2. Association between the Variables and the Perceptions Related to the Pandemic

The unadjusted analysis between demographic variables and professional and preventive knowledge about COVID-19 is reported in Table 3. Dental surgeons older than 60 years had no fear of becoming infected (OR 0.17; 95% CI 0.40–0.71); this association was statistically significant. Dentists who graduated from public institutions reported a fear of becoming infected when compared to professionals who studied dentistry at private institutions.

Dentists working in the public and private sectors who specialized in dentistry with workloads ranging from 11–20 h per week had a greater fear of becoming infected compared to other professionals (Table 3).

In the adjusted model, dentists practicing in the public sector and those working in both sectors had a greater fear of becoming infected with COVID-19 when compared with professionals who only operated a private practice (OR 2.46, 95% CI 1.02–5.45; and 3.81, CI 1.59–9.16) (Table 4). Dentists with a specialization had no fear of becoming infected with COVID-19.

## 4. Discussion

The present study aimed to investigate psychosocial factors of Brazilian dentists facing the risk of COVID-19 infection in the public and private sectors. Professionals working in public service presented greater psychosocial changes, e.g., fear and anxiety, regarding the risk of contracting COVID-19. Moreover, being a specialist was a protective factor against the challenge of working during the pandemic. During the performance of dental activities, the presence of body fluids that can be contaminated with SARS-CoV-2 and the use of equipment that produces aerosols directly in the oral cavity increase the probability of dentists becoming infected with COVID-19 [12,13,14]. This scenario has been impacting the work and emotional state of these professionals [15]. Two-third of dental professionals reported a fear of contracting COVID-19 while performing their duties and acting as vectors of viral transmission to their families, and more than half reported being afraid of dying from COVID-19 infection. Such numbers highlight the strong impacts of the pandemic on emotional health. These data are in agreement with those found by Ahamed et al. (2020), who, through an online questionnaire of professionals from 30 countries found that more than two-thirds of dentists were afraid of contracting COVID-19 during their work activities and 92% were scared of contracting the virus and infecting their family members [16]. The professionals in the age group above 60 years were less influenced by psychosocial factors since this population group had already received the doses of the vaccine against COVID-19, which resulted in greater confidence in their professional performance. Equivalent results were found in a study in Poland, where 80.7% of dental professionals feared contracting the disease at work and 90.1% were afraid of transmitting the virus to their family members [17]. Another study conducted in Ireland reported that 20% of dentists had stopped their activities (temporarily or permanently) and 47% of those answering questionnaires agreed that the pandemic was responsible for a financial loss of almost 90% [18]. These data are in agreement with those published by the British Dental Association (BDA) in which British dentists experienced a considerable financial decline due to the suspension of routine dental care [19]. In Brazil, it was no different as professionals who worked only in private clinics felt there had been a significant reduction in their monthly earnings. In the adjusted analysis, working in the SUS or in both types of service was statistically associated with a greater fear of contamination, which can be justified by the different contexts in which these professionals are inserted. In Brazil, the health care and oral health care model is universal, provided within the Brazil Unified Health System (SUS). (author). Public dental services are offered in primary care through the oral health teams in the Family Health Strategy, Dental Specialties Centers (CEOs), dental laboratories, and hospitals [20].

The increase in the number of people infected with COVID-19 has led to a greater demand for primary care, e.g., more equipment, materials, and human resources; as a result, dental surgeons found themselves situated on the front line with other health professionals [21] where they were reallocated to fast-track services, to perform rapid tests, and to collect biological material via swabs [22]. According to Santos et al. (2020) this new health scenario brought about a sense of fear, insecurity, and uncertainty to professionals [23]. Souza (2021) also added that at the time of the pandemic, primary care professionals had heavier workloads, leading to stress and, as a consequence, fear and less engagement at work. In addition, new biosafety protocols were enacted and adjusted according to the evolution of scientific knowledge concerning the risks of COVID-19 infection [24].

The biggest concerns reported by workers during care were the scarcity of personal protective equipment (PPE) and the lack of knowledge needed to meet demands, both of which contributed to clinical manifestations, increased stress, and insecurity, which support the results of this study [15].

Regarding the perceptions of work caused by the pandemic among these professionals, we found that 77.5% wanted to continue in the profession, 13.9% had reduced their working hours to be less exposed to the contagion, 3.3% had stopped seeing patients who needed invasive procedures, 4% did not want to leave their jobs in dentistry, but now conducted non-clinical activities, e.g., research or administrative functions, and only 1.3% did not intend to continue in the profession. Data that corroborate scientific evidence [25], where the results suggest that health professionals who work on the front line are more prone to mental disorders such as anxiety, depression, post-traumatic stress disorder, and sleep issues, which have a greater impact on these professionals’ quality of life. Future studies should be conducted to better understand these findings.

It is worth mentioning that the data from this study were collected at the end of 2020, the period following the peak of the first wave of infection and deaths in Brazil [26], which may have contributed to professionals reporting greater stress about infection in the workplace, fear of complications and deaths caused by SARS-CoV-2, as well as the possibility of being the vector of transmission to their families. Although these professionals had already received the second dose of the vaccines available in the country, many of their family members had not even received the first dose, which may have caused lingering stress and fear of infection among these professionals.

Some limitations should be noted in the present study, such as limitations of causal inference and sample. Specifically, despite the significant number of dentists who participated in the research, the sample should be reanalyzed with certain criteria since the participants were invited through the researchers’ social networks. That said, the sample size was adequate to explain the predictor variables with robust data. Therefore, to better understand these relationships, longitudinal studies should be conducted and other factors related to professionals’ quality of life should be evaluated. Studies comparing the influence of the work environment on professional performance are scarce. Another limitation of this study is the use of closed instruments to measure participants’ perceptions, which can prevent the knowledge of the circumstances and important details that would improve the quality of the answers. Accordingly, qualitative studies on the subject should be encouraged.

As for the strengths of this study, we highlight the need for more support in the psychosocial field, which would allow dentists to overcome difficulties and maintain good dental care services for the population. Continuing education should update professionals on the requisite scientific and clinical knowledge required to face the pandemic and to stimulate greater reflection on their role within this new context, which can improve their professional and emotional performance.

## 5. Conclusions

This research allows us to affirm that dentists who work in the public or public/private service experience fear and anxiety about becoming infected by COVID-19 during patient care.

## Figures and Tables

**Table 1 ijerph-19-10576-t001:** Descriptive analysis of sociodemographic variables and professional variables.

Sociodemographic Variables	n (%)
Sex	
Male	88 (29.2%)
Female	214 (70.8%)
Age group	
20–30 years	54 (17.9%)
31–40 years	65 (21.5%)
41–50 years	112 (37.1%)
51–60 years	61 (20.2%)
>60 years	10 (3.3%)
Professional variables	
College Type	
Private	117 (38.7%)
Public	185 (61.3%)
Years of graduation	
0–5 years	50 (16.6%)
6–10 years	41 (13.6%)
11–15 years	29 (9.6%)
16–20 years	34 (13.3%)
21–25 years	42 (13.9%)
26–30 years	59 (19.5%)
30 years	47 (15.6%)
Service Type	
Private	46 (15.3%)
Public	199 (66.1%)
Both	56 (18.6%)
Clinical performance	
Clinician	130 (43%)
Specialist	172 (57%)
Workload	
1–10 h	36 (11.9%)
11–20 h	30 (9.9%)
21–30 h	45 (14.9%)
31–40 h	98 (32.5%)
41–50 h	71 (23.5%)
More than 50 h	22 (7.3%)

Source: The author.

**Table 2 ijerph-19-10576-t002:** Dentists’ perceptions of the COVID-19 pandemic.

Dentists Perceptions	Yes	No	Sometimes/Maybe
Are you afraid of being infected with COVID-19 while performing a dental procedure?	226 (75%)	76 (25%)	
Are you afraid of becoming infected during your work and transmitting the virus to your family members?	260 (86%)	42 (14%)	
Are you afraid when you hear that people are dying from COVID-19?	163 (54%)	44 (16%)	95 (30%)
Do you think you have the necessary protective knowledge regarding COVID-19?	193 (64%)	35 (12%)	74 (24%)

Source: The author.

**Table 3 ijerph-19-10576-t003:** Unadjusted logistic regression among factors associated with fear of becoming infected by COVID-19.

Fear of Becoming Infected with COVID-19			
Variables	OR	IC 95%	*p* Value
Demographic			
Sex			
Male	1		
Female	0.92	0.52–1.65	0.8
Age Group			
20–30 years old	1		
31–40 years old	0.96	0.38–2.26	0.87
41–50 years old	0.66	0.30–1.45	0.31
51–60 years	0.94	0.38–2.32	0.9
>60 years	0.17	0.40–0.71	0.01
Professional Variables			
College Type			
Private	1		
Public	1.16	0.67–2.00	0.57
Time of graduation			
0–5 years	1		
6–10 years	1.84	0.62–5.43	0.26
11–15 years	1.51	0.47–4.84	0.48
16–20 years	0.63	0.24–1.63	0.34
21–25 years	0.88	0.34–2.29	0.8
26–30 years	0.96	0.39–2.35	0.94
>30 years	0.67	0.27–1.64	0.38
Service Type			
Private	1		
Public	2.46	1.04–5.81	0.04
Both	3.15	1.35–7.36	<0.00
Clinical Performance			
Clinician	1		
Specialist	1.4	1.04–1.88	0.05
Workload			
1–10 h	1		
11–20 h	1.87	0.51–6.77	0.33
21–30 h	1.23	0.34–4.3	0.74
31–40 h	1.15	0.36–3.69	0.8
41–50 h	1.29	0.36–3.69	0.62
50+ hours	0.78	0.27–2.26	0.65
Perception of COVID-19			
Preventive knowledge			
No			
Yes	1		
Maybe	0.32	(0.1–0.97)	0.04
	0.51	(0.15–1.69)	0.27

Source: The author. OR: odds ratio; 95% CI: 95% confidence interval; *p* < 0.20.

**Table 4 ijerph-19-10576-t004:** Adjusted multivariate logistic regression between factors associated with fear of becoming infected.

Fear of Becoming Infected with COVID-19			
Variables	OR	IC 95%	*p* Value
**Demographic**			
Age group			
20–30 years old	1		
31–40 years old	0.70	0.20–2.42	0.57
41–50 years old	0.61	0.14–2.60	0.51
51–60 years	0.79	0.14–4.35	0.79
>60 years	0.15	0.19–1.26	0.08
**Occupational Variables**			
Work sector			
Private	1		
Public	2.70	1.06–6.87	0.04
Both	3.21	1.34–7.69	<0.00
Clinical practice			
Clinician	1		
Specialist	0.55	0.30–1.01	0.05
**Clinical Acting**			
Clinical	1		
Specialist	0.55	0.30–1.01	0.05
**Perception of COVID-19**			
Preventive knowledge			
No			
Yes	1		
Maybe	2.35	0.74–7.4	0.14
	1.29	0.65–2.58	0.62

Source: The author. OR: odds ratio, 95% CI: 95% confidence interval; *p* < 0.05.

## Data Availability

The data presented in this study are available on request from the corresponding author.

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
