# Peer review of "Assessment of Psychosocial Factors in Brazilian Dentists Facing Risk of SARS-CoV-2 Infection in the Public and Private Sectors"

_ijerph, 2022, doi:10.3390/ijerph191710576_

Round 1

Reviewer 1 Report

I want to thank the authors for emphasizing care for mental health during covid in one of the most risk professions for disease transmission. 

In introduction in chapter starting with line 45, dental surgeons were  not at the risk as other general dentists because of aerosol producing devices, but because they were the profession that was dealing with all the emergencies in the dental profession in the begining of the pandemic. 

There should be an explanation in the Materials and method section, why the authors have chosen oral surgeons as targeted subjects?

Lines 109-114 were repeating. 

How was Perception to coping with the COVID-19 Pandemic were estimated.?

Lines 127-142. Authors are repeating the results presented in the tables also in the text. Please correct it. it is well non rule in scientific writing that tables are not to be repeated with the text. 

Lines 156-162 are explanations that do belong in the Discussion section.

In table 3 there are two types of profession, specialist and clinician. What is the difference between the two? as stated before, all are oral surgeons., or there are also some other type of specialists? 

At line 177 there is a sentence: Dentists who had specialization had no fear of becoming infected with 177 COVID-19. Is there a possibility that a dentist without specialization works as oral surgeon?

The last sentence  (recommendation) in the discussion : As strong points of this study, we highlight the need for more support in the psychosocial field... should be also addressed in the last sentence of summary. 

Author Response

Dear Reviewer,

Reviewer 2 Report

Great and interesting study but the limitations were relevant due to the recruitment process.

A quantitative study has limitations so the authors did use closed ended questions which are not strong enough to provide the evidence-based that can enhance the results.

it is important to provide more details of recognizing such weaknesses.

and provide recommendations to engage with a qualitative design to really explore these psychosocial factors.

Author Response

Good morning, thank you for the considerations

Please see the attached document with the corrections.

Yours sincerely

Round 2

Reviewer 1 Report

The authors did improve the manuscript by the suggestions from both reviewers. 

One major issue still needs to be solved. The conclusion has to be the answer to the aim of the study, or hypothesis of the study, which is not the case in this manuscript. Please correct this. 

aim: Accordingly, the present study aimed to investigate the perceptions of dental professionals working in the public and private sectors regarding fear and anxiety in patient care and the risk of infection in the context of the COVID-19 pandemic.

answer (conclusion): This research allows us to affirm that dentists who work in public or public/private service are more prone to suffer alterations in psychosocial factors and face the risk of COVID-19 infection, which can harm to their quality of life.

Author Response

The conclusion is corrected.

This research allows us to affirm that dentists who work in the public or public/private service have fear and anxiety about being infected by covid-19 in patient care.

Sincerely,

Gabriela Meira